# A machine learning approach identifies distinct early-symptom cluster phenotypes which correlate with hospitalization, failure to return to activities, and prolonged COVID-19 symptoms

Nusrat J. Epsi[1,2], John H. Powers[3], David A. Lindholm[4,5], Katrin Mende[1,2,4], Allison Malloy[6], Anuradha Ganesan[1,2,7], Nikhil Huprikar[7], Tahaniyat Lalani[1,2,8], Alfred Smith[8], Rupal M. Mody[9], Milissa U. Jones[10], Samantha E. Bazan[11], Rhonda E. Colombo[1,2,5,12], Christopher J. Colombo[12], Evan C. Ewers[13], Derek T. Larson[5,13,14], Catherine M. Berjohn[1,5,14], Carlos J. Maldonado[15], Paul W. Blair[2,16], Josh Chenoweth[2], David L. Saunders[17], Jeffrey Livezey[17], Ryan C. Maves[18], Margaret Sanchez Edwards[1,2], Julia S. Rozman[1,2], Mark P. Simons[1], David R. Tribble[1], Brian K. Agan[1,2], Timothy H. Burgess[1], Simon D. Pollett[1,2], Stephanie A. Richard[1,2]* for the EPICC COVID-19 Cohort Study Group**

**1** Infectious Disease Clinical Research Program, Department of Preventive Medicine and Biostatistics, Uniformed Services University of the Health Sciences, Bethesda, Maryland, United States of America, **2** Henry M. Jackson Foundation for the Advancement of Military Medicine, Inc., Bethesda, Maryland, United States of America, **3** Clinical Research Directorate, Frederick National Laboratory for Cancer Research, Frederick, Maryland, United States of America, **4** Molecular Biology Laboratory, Brooke Army Medical Center, San Antonio, Texas, United States of America, **5** Department of Medicine, Uniformed Services University of the Health Sciences, Bethesda, Maryland, United States of America, **6** Department of Pediatrics, Walter Reed National Military Medical Center, Bethesda, Maryland, United States of America, **7** Infectious Disease Clinic, Walter Reed National Military Medical Center, Bethesda, Maryland, United States of America, **8** Infectious Disease Clinical Research Program, Naval Medical Center Portsmouth, Portsmouth, Virginia, United States of America, **9** Infectious Disease Clinic, William Beaumont Army Medical Center, El Paso, Texas, United States of America, **10** Pediatric Infectious Diseases, Tripler Army Medical Center, Honolulu, Hawaii, United States of America, **11** Family Nurse Practitioner and Women's Health Nurse Practitioner Program, Carl R. Darnall Army Medical Center, Fort Hood, Texas, United States of America, **12** Infectious Disease Clinic, Madigan Army Medical Center, Tacoma, Washington, United States of America, **13** Internal Medicine, Fort Belvoir Community Hospital, Fort Belvoir, Virginia, United States of America, **14** Infectious Disease Clinic, Naval Medical Center San Diego, San Diego, California, United States of America, **15** Department of Research and Clinical Investigation, Womack Army Medical Center, Fort Bragg, North Carolina, United States of America, **16** Department of Pathology, Uniformed Services University of the Health Sciences, Bethesda, Maryland, United States of America, **17** Translational Medicine Unit, Department of Medicine, Uniformed Services University of the Health Sciences, Bethesda, Maryland, United States of America, **18** Infectious Diseases and Critical Care Medicine, Wake Forest School of Medicine, Winston-Salem, North Carolina, United States of America

** Membership of the EPICC COVID-19 Cohort Study Group is provided in the Acknowledgments.
* srichard@idcrp.org

## Abstract

### Background

Accurate COVID-19 prognosis is a critical aspect of acute and long-term clinical management. We identified discrete clusters of early stage-symptoms which may delineate groups

the Creative Commons CC0 public domain dedication.

**Data availability statement:** All relevant data are provided within the paper and its Supporting Information files.

**Funding:** This work was supported by awards from the Defense Health Program (HU00012020067) and the National Institute of Allergy and Infectious Disease (HU00011920111). The protocol was executed by the Infectious Disease Clinical Research Program (IDCRP), a Department of Defense (DoD) program executed by the Uniformed Services University of the Health Sciences (USUHS) through a cooperative agreement by the Henry M. Jackson Foundation for the Advancement of Military Medicine, Inc. (HJF). This project has been funded in part by the National Institute of Allergy and Infectious Diseases at the National Institutes of Health, under an interagency agreement (Y1-AI-5072).

**Competing interests:** Potential conflicts of interest. S. D. P., T. H. B, D.R.T, and M.P.S. report that the Uniformed Services University (USU) Infectious Diseases Clinical Research Program (IDCRP), a US Department of Defense institution, and the Henry M. Jackson Foundation (HJF) were funded under a Cooperative Research and Development Agreement to conduct an unrelated phase III COVID-19 monoclonal antibody immunoprophylaxis trial sponsored by AstraZeneca. The HJF, in support of the USU IDCRP, was funded by the Department of Defense Joint Program Executive Office for Chemical, Biological, Radiological, and Nuclear Defense to augment the conduct of an unrelated phase III vaccine trial sponsored by AstraZeneca. Both of these trials were part of the US Government COVID-19 response. Neither is related to the work presented here. This does not alter our adherence to PLOS ONE policies on sharing data and materials.

with distinct disease severity phenotypes, including risk of developing long-term symptoms and associated inflammatory profiles.

## Methods

1,273 SARS-CoV-2 positive U.S. Military Health System beneficiaries with quantitative symptom scores (FLU-PRO Plus) were included in this analysis. We employed machine-learning approaches to identify symptom clusters and compared risk of hospitalization, long-term symptoms, as well as peak CRP and IL-6 concentrations.

## Results

We identified three distinct clusters of participants based on their FLU-PRO Plus symptoms: cluster 1 ("Nasal cluster") is highly correlated with reporting runny/stuffy nose and sneezing, cluster 2 ("Sensory cluster") is highly correlated with loss of smell or taste, and cluster 3 ("Respiratory/Systemic cluster") is highly correlated with the respiratory (cough, trouble breathing, among others) and systemic (body aches, chills, among others) domain symptoms. Participants in the Respiratory/Systemic cluster were twice as likely as those in the Nasal cluster to have been hospitalized, and 1.5 times as likely to report that they had not returned-to-activities, which remained significant after controlling for confounding covariates ($P < 0.01$). Respiratory/Systemic and Sensory clusters were more likely to have symptoms at six-months post-symptom-onset ($P = 0.03$). We observed higher peak CRP and IL-6 in the Respiratory/Systemic cluster ($P < 0.01$).

## Conclusions

We identified early symptom profiles potentially associated with hospitalization, return-to-activities, long-term symptoms, and inflammatory profiles. These findings may assist in patient prognosis, including prediction of long COVID risk.

## Introduction

The SARS-CoV-2 pandemic continues to burden the healthcare system, and the clinical spectrum of Coronavirus disease 2019 (COVID-19) ranges from asymptomatic to severe or critical illness [1]. Older age and medical comorbidities have been associated with a higher risk for severe COVID-19 outcomes [1–3]. In addition to variability in acute illness severity, duration of illness can vary across individuals, with many recovering within several weeks, and others reporting symptoms for months, a phenomenon termed Post-COVID conditions (PCC, or "Long COVID") [4]. Individuals with Long COVID exhibit a wide variety of symptoms, including loss of sense of taste and smell, fatigue, dyspnea, arthralgia, chest pain, myalgia, and cough [5–11].

Predicting such long-term outcomes after an initial COVID-19 illness is a priority. Due to the broad case definition, inconsistent self-reporting, and the non-specific nature of frequently observed symptoms, predictions based on acute clinical presentation remains elusive [12]. Emerging research has focused on early biomarker signatures, including immune responses and acute imaging [13–15], but these are not widely accessible in routine care. Moreover, they are often focused on populations requiring hospitalization for COVID-19, rather than

patients who are treated in outpatient settings but nonetheless still carry a risk of long-term sequelae even after vaccination [16]. While some studies have identified acute symptom clusters [17–19], they have not yet fully explored the biological disease phenotype association with symptoms within each cluster, and/or have focused on severe acute COVID-19 rather than chronic outcomes [17, 20]. Further, acute symptom data used in prognostic studies are often not measured using validated patient symptom scoring systems, which are critical given the inherent subjectivity and variability in elicited symptoms [18–20].

In this work, we sought to group early COVID-19 symptoms using machine learning techniques and describe the relationships between those acute symptom groups and short- and long-term clinical outcomes of SARS-CoV-2 infection. We then compared inflammatory biomarkers among these clusters to further characterize their biological significance. We utilized InFLUenza Patient-Reported Outcome (FLU-PRO) Plus [12, 21], which is a standardized instrument designed to characterize the frequency, intensity, and duration of symptoms in viral respiratory infections, when administered serially over time. Specifically, we sought to *(i)* identify infected individuals who exhibit similar acute symptoms using machine learning methods and delineate symptom-based acute phenotypes with precision, *(ii)* evaluate the relationship between these acute symptom clusters and acute COVID-19 hospitalization status, *(iii)* evaluate the relationship between acute symptom cluster and reported return to usual activities and health, and *(iv)* evaluate the relationship between acute symptom clusters and COVID-19 symptoms at six months post-symptom onset, Finally, we *(v)* explored whether patients in different acute symptom clusters have different host inflammatory responses.

## Materials and methods

### Study population and general study design

The Epidemiology, Immunology, and Clinical Characteristics of Emerging Infectious Diseases with Pandemic Potential (EPICC) study is a longitudinal cohort study of U.S. Military Health System (MHS) beneficiaries designed to examine the clinical severity and long-term outcomes of SARS-CoV-2 infection [22]. Briefly, MHS beneficiaries presenting to one of ten participating military treatment facilities (MTFs) with confirmed COVID-19, a COVID-19-like illness, or a high-risk exposure to someone with COVID-19 were eligible for enrollment in EPICC. Later in the study, eligibility expanded to include an online component, in which individuals who were tested for or vaccinated against SARS-CoV-2 could also enroll. The participants included in this analysis were adults enrolled between March 20, 2020, and March 31, 2022, tested positive for SARS-CoV-2, and filled out at least one FLU-PRO Plus survey. We calculated the Charlson Comorbidity Index (CCI) [23] using documented health encounters in the MHS Data Repository (MDR). Body mass index (BMI) was calculated using height and weight values collected in the surveys and from the MDR and categorized as normal/underweight ($\leq$24.9 kg/m$^2$), overweight (25–29.9 kg/m$^2$), obese (30–34.9 kg/m$^2$), and severely obese ($\geq$35 kg/m$^2$). Age, sex, and race/ethnicity were reported by the participant. COVID-19 hospitalization was determined based on survey responses reporting hospitalization due to COVID-19 and case report forms filled out by study staff.

### FLU-PRO© Plus

The FLU-PRO© instrument was originally developed to assess influenza-like symptoms [24], and has since been evaluated for use in other respiratory infections [25, 26]. The original FLU-PRO instrument has been updated to include loss of sense of smell and taste (FLU-PRO Plus), and was found to have high reliability and construct validity for use in SARS-CoV-2 studies [12]. The FLU-PRO Plus instrument was implemented only in those

enrolled at an MTF (not in online participants) and includes questions about 34 symptoms that provide a direct measure of the presence and severity of symptoms across seven body systems termed "domains" (Nose, Throat, Eyes, Chest/Respiratory, Gastrointestinal, Body/Systemic, and Senses) using a 5-point ordinal severity scale. The responses ranged from "Not at all" to "Very much" for most questions, and "Never" to "Always" for sneezing, coughing, and coughed up mucus or phlegm, and the number of times (0 to 4 or more) for vomiting and diarrhea. Domain-specific scores, as well as a total score, are calculated using the mean of all symptoms within the domain/total. Participants completed the FLU-PRO Plus survey daily for 14 days, and we used the earliest response reported by each participant for this analysis. The FLU-PRO Plus survey also includes questions about whether the participant has returned to their usual health or activities, and for these questions, we considered whether they reported that they had returned to their usual health or activities at the time of their last FLU-PRO Plus.

### Diagnosis of SARS-CoV-2 infection and genotype

In EPICC, SARS-CoV-2 infection was determined by one of the following criteria: a positive clinical PCR test, PCR positive swab collected as part of the EPICC study, or report of a positive test by the participant. The PCR assay used in this study for study-collected swabs was the SARS-CoV-2 (2019-nCoV) Centers for Disease Control and Prevention (CDC) quantitative PCR (qPCR) Probe Assay (IDT, Coralville, IA). Nucleocapsid (N) genes (N1 and N2) were targeted in this assay, human RNase P gene (Rp) acts as a specimen control, and a cycle threshold (CT) less than 40 for N1 and N2 protein was considered positive for SARS-CoV-2 infection. To determine genotype, whole-genome sequencing of SARS-CoV-2 using 1200 bp tiled amplicons was applied to upper respiratory swabs [27]. Illumina Nextera®XT DNA Library Preparation Kit was used to prepare amplified product and the Pangolin lineage assignment tool was utilized to classify the genotype [28].

### Measurement of CRP and IL-6 levels

C-reactive protein (CRP) and Interleukin-6 (IL-6) were measured in the plasma samples using the high dynamic range enzyme-linked immunosorbent assay (ELISA) microfluidics analyzer (ProteinSimple, San Jose, California, USA). CRP and IL-6 were $log_{10}$ transformed and empirical Bayes frameworks were used to adjust data for batch effects. Data imputation was done using the $k$-nearest neighbour algorithm [29]. For this analysis we included plasma samples within 21 days post-symptom onset.

### Ascertainment and definitions of COVID-19 vaccination group

We obtained vaccination details from MDR records, case report forms, and surveys [30]. We identified participants as fully vaccinated if 14 or more days had passed since their second dose of an mRNA vaccine series (Pfizer/BioNTech-BNT162b2, Moderna mRNA-1273). Vaccine breakthroughs were identified if a participant tested positive for SARS-CoV-2 14 or more days after the final vaccine dose of the series.

### Identification of prolonged COVID-19 symptoms

Along with the FLU-PRO Plus survey, EPICC participants were also requested to complete online follow-up surveys that included questions about presence and severity of ongoing symptoms at 1, 3, 6, 9, and 12 months. Participants who responded at approximately six months post-symptom onset (135–224 days) were included in the prolonged COVID-19 symptoms analysis (S1 Fig in S1 File). If the participant reported continuing symptoms on

the follow-up survey, they were asked about specific symptoms (cough, dyspnea (difficulty breathing/shortness of breath), exercise intolerance, loss of sense of smell and/or taste, joint pain, fatigue, headache, etc.) and asked to rate the severity of those symptoms (none, mild, moderate, severe, and critical). Participants who reported any moderate to critical symptoms at six-months post-symptom onset were considered to have prolonged COVID-19 symptoms in this analysis.

## Cluster analysis

Rather than grouping symptoms based on *a priori* assumptions, we used machine learning clustering of FLU-PRO Plus responses to identify symptom patterns. Optimal clustering can be a subjective process which is dependent on the characteristics used for determining patterns of commonality and dissimilarity. We applied principal component analysis [31], which performs a linear mapping of the data to a lower dimension space in such a way that the variance of the data in the low-dimensional representation is maximized. It does so by calculating eigenvectors from the covariance matrix. The eigenvectors that correspond to the largest principal components are used to reconstruct a significant fraction of the variance of the original data. As an added benefit, each of the new features or components created after PCA are all independent of one another. Therefore to visualize the pattern with much greater granularity we applied unsupervised machine learning algorithm K-means [32] to view the top PCA components. To do that, first, we determine the number of clusters *k* by using statistical testing method gap statistics [33]. This method compares observed data and reference data with a random uniform distribution and identifies clusters by choosing the value that maximizes the gap. Maximum gap value signifies that the clustering structure is far away from the random uniform distribution of the data points. Therefore, PCA followed by K-means helps to identify groups with distinct clusters of symptoms. This algorithm clusters subjects so that symptoms that are highly correlated are clustered together. We further characterized the clusters using the mean domain response [24], and annotated each cluster with the predominant domain.

## Adjusted comparisons of clusters and acute and long-term outcomes

Univariable Poisson regression was performed to evaluate whether the identified clusters and other independent variables were associated with the outcomes of COVID-19, including hospitalization, return to usual activities, return to usual health, and prolonged COVID-19 symptoms. Multivariable Poisson regression was performed for each of the outcomes, adjusting for other factors including age group, sex, race/ethnicity, obesity, CCI category, vaccine receipt, and days post-symptom onset of the first FLU-PRO Plus survey. Adjusted risk ratios (aRR) and 95% confidence limits (CIs) were calculated. Model fit was estimated by the Akaike information criterion (AIC) and Bayesian information criterion (BIC).

We performed unadjusted and adjusted linear regression to quantify the relationship between identified clusters and participants' acute plasma inflammatory biomarker CRP and IL-6 levels. These models considered other potential predictors of clusters, including specimen sampling time since symptom onset, sex, age group, race/ethnicity, obesity, CCI category, and vaccine receipt.

## Statistical analysis

Descriptive statistics were calculated for the demographic characteristics, CCI category, BMI category, vaccination status, variants, vaccine receipt, return to usual activities, return to usual health, and days post-symptom onset, with *P* values computed using Fisher's exact test. Figures were generated and statistical analyses were performed in RStudio version 4·0·2 [34].

### Ethics

This study was approved by the Uniformed Services University of the Health Sciences (USUHS) Institutional Review Board (IRB) under protocol IDCRP-085 [22]; all participants or their legally authorized representative provided informed consent to participate.

## Results

### Demographic and clinical characteristics

Among 2552 participants enrolled in EPICC at an MTF from March 2020 through March 2022, 2407 were SARS-CoV-2 positive, and 1273 responded to their first FLU-PRO Plus survey within 21 days post-symptom onset (S1 Fig and S1 Table in S1 File). The responder study sample was predominately male (58.3%), 18–44 years of age (60.2%), and had no comorbidities at enrollment (62.3% had a CCI score of zero, Table 1). About one in five participants in this analysis were hospitalized due to COVID-19 (19.1%). In the non-responder group (those who did not filled out FLU-PRO Plus survey, N = 433), which was excluded from the analysis, consisted of approximately 20% children, 29.3% hospitalized individuals, and 13.7% tested negative for SARS-CoV-2. Only 2 participants reported moderate to severe symptoms at 6 months, and overall, the group dominantly exhibits dependent (42.0%) than the responder group (S1 Table in S1 File).

### Acute COVID-19 symptoms group together in three distinct clusters

To define distinct early-stage symptom profiles, we employed principal component analysis followed by the k-means clustering technique (Fig 1A–1C). Cluster 1 exhibited a higher mean score of nasal symptoms (e.g., runny or stuffy nose), thus termed the 'Nasal cluster'. Cluster 2 exhibited a higher mean score of sensory symptoms (e.g., loss of sense of smell or taste), thus is the 'Sensory cluster'. Cluster 3 exhibited a higher mean score of respiratory (e.g., upper and lower respiratory) and systemic symptoms (e.g., body ache, chills), annotated as the 'Respiratory/Systemic cluster' (Fig 1D).

The Respiratory/Systemic cluster of cases had a higher proportion of participants that were hospitalized (36.3%) and had more comorbidities (46.6% with CCI > 0) than the Nasal (hospitalized: 11.9%; CCI > 0: 39.5%) or Sensory (hospitalized: 10.5%; CCI >0: 28.1%) clusters (*P* < 0.01); those participants in the Sensory cluster appeared to be younger than the other clusters (70.2% were 18–44 years old), compared to Nasal (59.1%) and Respiratory/Systemic (50.1%) clusters (*P* < 0.01) (Table 1). Those with Nasal cluster symptom profiles corresponded with a higher proportion of Omicron variant infections (BA.1/BA.2) (13.3%) compared to Sensory (2.5%) and Respiratory/Systemic (4.5%) clusters. Nasal cluster acute symptom profiles were more likely to be associated with vaccine breakthrough cases (37.4%) compared to Sensory (20.4%) and Respiratory/Systemic (19.3%) clusters. Self-reported return to usual activities and health at their last FLU-PRO Plus survey was more common in those with Nasal cluster symptoms compared to the other clusters (Table 1). In the prolonged COVID-19 subset, the Nasal cluster had a lower proportion reporting symptoms at six months (3.8%) than the other two clusters (Sensory: 12%; Respiratory/Systemic: 11.2%) (Table 1).

### Acute COVID-19 symptom profiles defined by machine learning are independently associated with hospitalization and long-term symptom persistence risk

The Respiratory/Systemic cluster was associated with more than a two-fold (aRR = 2.24 [95% CI: 1.61 to 3.12], *P* < 0.01) increased risk of hospitalization compared to the Nasal cluster, after controlling for sex, age group, race/ethnicity, CCI category, BMI category, vaccine receipt, and days post-symptom onset (Fig 2 and S2 Table in S1 File). Older age, CCI category, and BMI

**Table 1. Clinical and demographic characteristics of 1273 military health system beneficiaries by early FLU-PRO symptom clusters.**

| | Nasal cluster[a] (N = 428) | Sensory cluster[b] (N = 446) | Respiratory/ Systemic cluster[c] (N = 399) | Total (N = 1273) | P value[d] |
|---|---|---|---|---|---|
| **Age group** | | | | | < 0.01 |
| 18–44 | 253 (59.1%) | 313 (70.2%) | 200 (50.1%) | 766 (60.2%) | |
| 45–64 | 121 (28.3%) | 110 (24.7%) | 153 (38.3%) | 384 (30.2%) | |
| 65+ | 54 (12.6%) | 23 (5.2%) | 46 (11.5%) | 123 (9.7%) | |
| **Sex** | | | | | < 0.01 |
| Female | 161 (37.6%) | 222 (49.8%) | 148 (37.1%) | 531 (41.7%) | |
| Male | 267 (62.4%) | 224 (50.2%) | 251 (62.9%) | 742 (58.3%) | |
| **Race/Ethnicity** | | | | | < 0.01 |
| Black | 51 (11.9%) | 54 (12.1%) | 53 (13.3%) | 158 (12.4%) | |
| Hispanic or Latino | 83 (19.4%) | 131 (29.4%) | 119 (29.8%) | 333 (26.2%) | |
| Other | 40 (9.3%) | 41 (9.2%) | 44 (11.0%) | 125 (9.8%) | |
| White | 254 (59.3%) | 220 (49.3%) | 183 (45.9%) | 657 (51.6%) | |
| **Charlson Comorbidity Index** | | | | | < 0.01 |
| 0 | 259 (60.5%) | 321 (72.0%) | 213 (53.4%) | 793 (62.3%) | |
| 1–2 | 99 (23.1%) | 82 (18.4%) | 121 (30.3%) | 302 (23.7%) | |
| 3–4 | 44 (10.3%) | 27 (6.1%) | 36 (9.0%) | 107 (8.4%) | |
| >5 | 26 (6.1%) | 16 (3.6%) | 29 (7.3%) | 71 (5.6%) | |
| **Severity** | | | | | < 0.01 |
| Hospitalized | 51 (11.9%) | 47 (10.5%) | 145 (36.3%) | 243 (19.1%) | |
| Outpatient | 377 (88.1%) | 399 (89.5%) | 254 (63.7%) | 1030 (80.9%) | |
| **Variants** | | | | | < 0.01 |
| Alpha | 11 (2.6%) | 11 (2.5%) | 16 (4.0%) | 38 (3.0%) | |
| Delta | 47 (11.0%) | 64 (14.3%) | 48 (12.0%) | 159 (12.5%) | |
| Omicron | 57 (13.3%) | 11 (2.5%) | 18 (4.5%) | 86 (6.8%) | |
| Other[e] | 160 (37.4%) | 211 (47.3%) | 180 (45.1%) | 551 (43.3%) | |
| Unknown[f] | 153 (35.7%) | 149 (33.4%) | 137 (34.3%) | 439 (34.5%) | |
| **Body Mass Index** | | | | | < 0.01 |
| Normal | 103 (24.1%) | 102 (22.9%) | 52 (13.0%) | 257 (20.2%) | |
| Overweight | 160 (37.4%) | 177 (39.7%) | 151 (37.8%) | 488 (38.3%) | |
| Obese | 89 (20.8%) | 106 (23.8%) | 118 (29.6%) | 313 (24.6%) | |
| Severely Obese | 76 (17.8%) | 61 (13.7%) | 78 (19.5%) | 215 (16.9%) | |
| **Days post-symptom onset of first FLU-PRO Plus survey** | | | | | < 0.01 |
| Median (Q1—Q3) | 10 (7.0–15.0) | 9 (7.0–13.0) | 11 (7.0–16.0) | 10 (7.0 to 14.0) | |
| **Vaccine breakthrough cases** | | | | | < 0.01 |
| No | 268 (62.6%) | 355 (79.6%) | 322 (80.7%) | 945 (74.2%) | |
| Yes | 160 (37.4%) | 91 (20.4%) | 77 (19.3%) | 328 (25.8%) | |
| **Military Status** | | | | | < 0.01 |
| Active duty | 214 (50.0%) | 254 (57.0%) | 166 (41.6%) | 634 (49.8%) | |
| Dependent | 116 (27.1%) | 133 (29.8%) | 95 (23.8%) | 344 (27.0%) | |
| Retired military | 98 (22.9%) | 59 (13.2%) | 138 (34.6%) | 295 (23.2%) | |
| **Department of Defense affiliation[g]** | | | | | 0.01 |
| Army | 201 (47.0%) | 214 (48.0%) | 170 (42.6%) | 585 (46.0%) | |
| Navy | 105 (24.5%) | 144 (32.3%) | 109 (27.3%) | 358 (28.1%) | |
| Air Force | 77 (18.0%) | 54 (12.1%) | 86 (21.6%) | 217 (17.0%) | |
| Marines | 30 (7.0%) | 23 (5.2%) | 24 (6.0%) | 77 (6.0%) | |
| Coast Guard | 3 (0.7%) | 3 (0.7%) | 5 (1.3%) | 11 (0.9%) | |
| Other | 12 (2.8%) | 8 (1.8%) | 5 (1.3%) | 25 (2.0%) | |

(*Continued*)

**Table 1.** (Continued)

| | Nasal cluster[a] (N = 428) | Sensory cluster[b] (N = 446) | Respiratory/ Systemic cluster[c] (N = 399) | Total (N = 1273) | P value[d] |
|---|---|---|---|---|---|
| **Return to Activities** | | | | | < 0.01 |
| No | 99 (23.1%) | 134 (30.0%) | 169 (42.4%) | 402 (31.6%) | |
| Yes | 329 (76.9%) | 312 (70.0%) | 230 (57.6%) | 871 (68.4%) | |
| **Return to Health** | | | | | < 0.01 |
| No | 163 (38.1%) | 202 (45.3%) | 209 (52.4%) | 574 (45.1%) | |
| Yes | 265 (61.9%) | 244 (54.7%) | 190 (47.6%) | 699 (54.9%) | |
| **Vaccine** | | | | | < 0.01 |
| Pfizer | 142 (33.2%) | 91 (20.4%) | 71 (17.8%) | 304 (23.9%) | |
| Moderna | 28 (6.5%) | 12 (2.7%) | 15 (3.8%) | 55 (4.3%) | |
| Unvaccinated | 258 (60.3%) | 343 (76.9%) | 313 (78.4%) | 914 (71.8%) | |
| **Moderate to severe symptoms at six-months (N = 529)[h]** | | | | | 0.02 |
| No | 154 (96.2%) | 176 (88.0%) | 150 (88.8%) | 480 (90.7%) | |
| Yes | 6 (3.8%) | 24 (12.0%) | 19 (11.2%) | 49 (9.3%) | |

[a]Nasal cluster exhibits a higher mean score of nasal symptoms (e.g., runny or stuffy nose)

[b]Sensory cluster exhibits higher mean score of sensory symptoms (e.g., loss of sense of smell or taste)

[c]Respiratory/System Cluster exhibits higher mean score of respiratory (e.g., upper and lower respiratory) and systemic symptoms (e.g., body ache, chills), respectively

[d]n x k Fisher's exact test

[e]Other variants: Beta, Eta, Epsilon, Gamma, Iota, Theta

[f] Sequence failed to be mapped to a lineage

[g]Other DOD affiliation includes National Guard, National Oceanic and Atmospheric Administration, US Public Health Service, and missing affiliations.

[h] Subset of participants who filled out surveys at six-months post-symptom onset

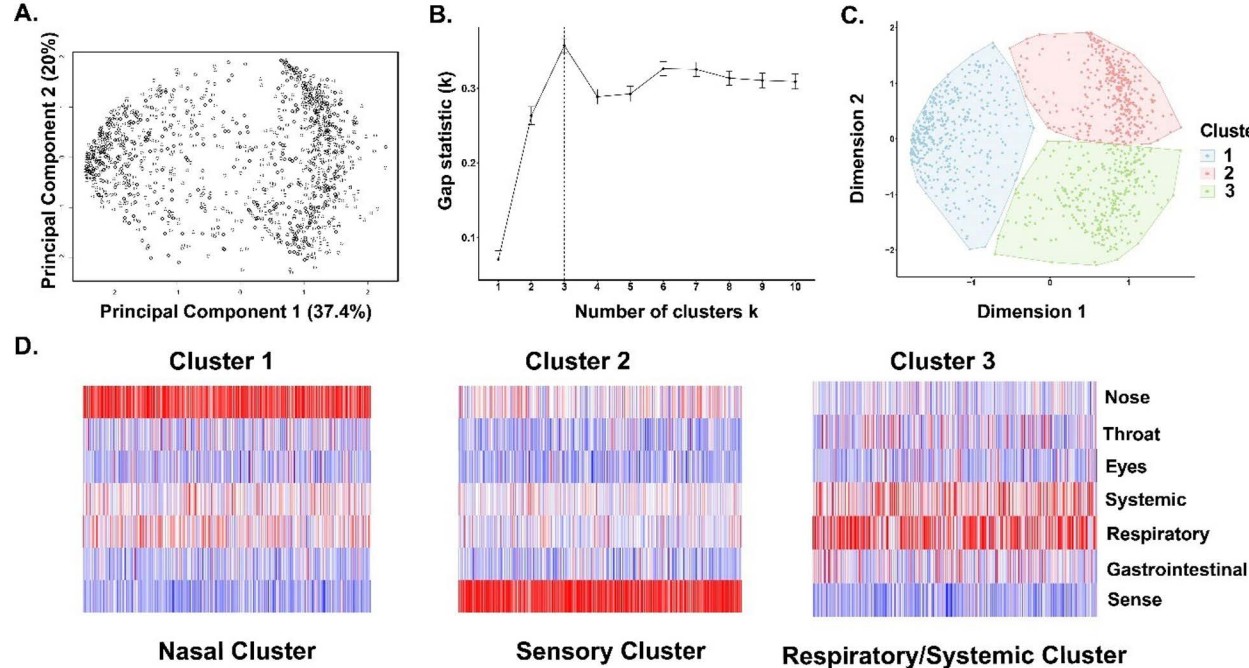

**Fig 1.** (A) Principal component analysis depicting FLU-PRO Plus response, (B) Optimal number of clusters using Gap-statistics, (C) K-means clustering identified three distinct clusters of participants, (D) Heatmap depicting three distinct clusters (high values are in red and low values are in blue).

category were also independently associated with a higher risk of hospitalization, after adjusting for all variables simultaneously. Compared to study participants in the Nasal cluster, those in the Respiratory/Systemic were more likely to report that they had not yet returned to activities or usual health on their last day with FLU-PRO Plus data (Fig 2 and S3, S4 Tables in S1 File).

Next, we evaluated the subset of 529 participants who filled out surveys at six-months post-symptom onset (S5 Table in S1 File). The most common symptoms reported at six months were fatigue (4.2%), loss of sense of smell or taste (4.0%), dyspnea (3.8%), and exercise intolerance (3.4%) (S6 Table in S1 File). We observed that those cases with acute symptom profiles belonging to the Sensory and Respiratory/Systemic clusters were more likely to report moderate to severe symptoms at six months than those belonging to the Nasal cluster (Sensory cluster: aRR = 2.86 [95% CI = 1.14 to 7.18], $P = 0.03$; Respiratory/Systemic cluster: aRR = 2.89 [95% CI = 1.12 to 7.44], $P = 0.03$) (S7 Table in S1 File).

## Acute COVID-19 symptom profile clusters have distinct inflammatory profiles

CRP and IL-6 levels are found to be higher in the Respiratory/Systemic cluster compared to the Nasal cluster after adjusting for age, sex, race/ethnicity, CCI category, BMI category,

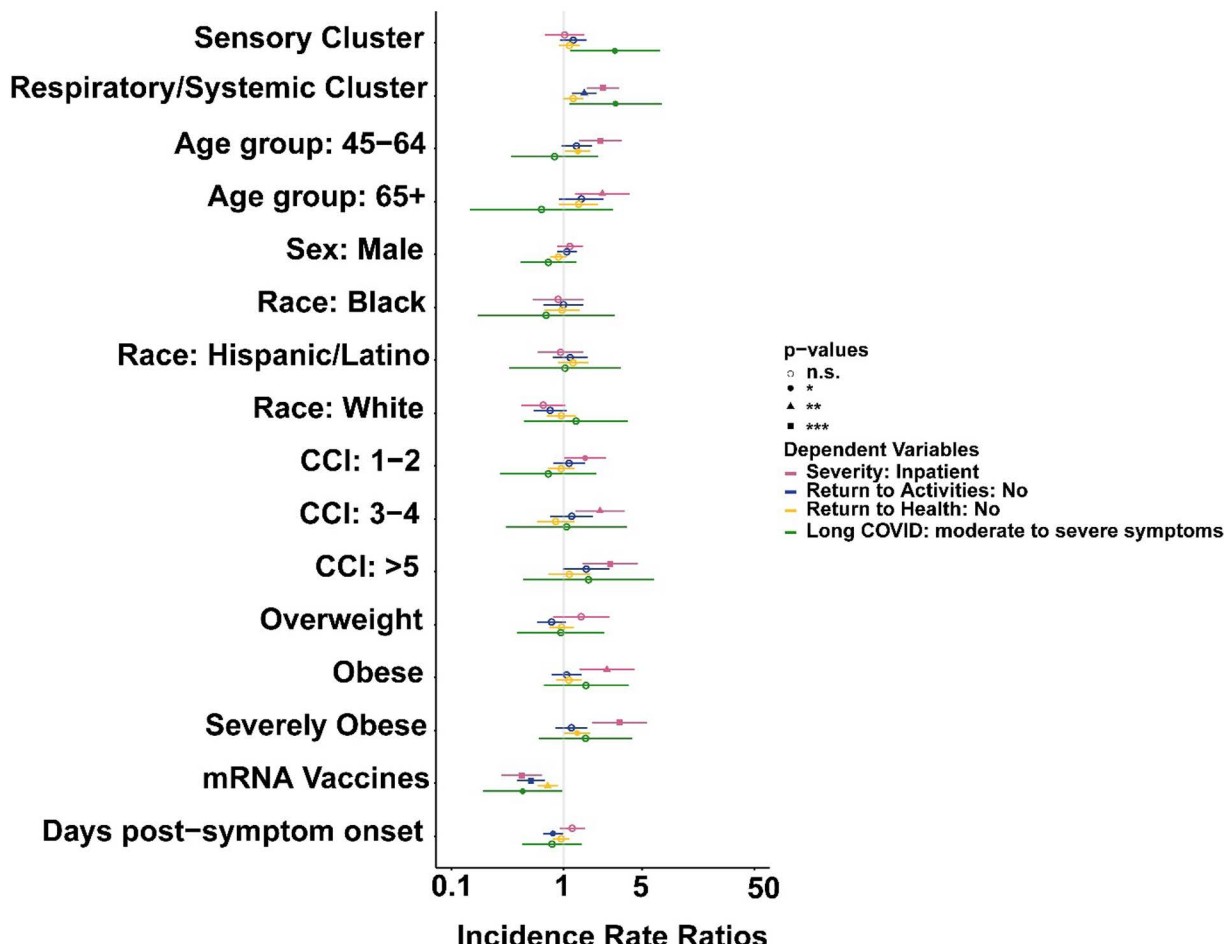

**Fig 2. Multivariable Poisson regression model results from four distinct models: Disease severity (*pink*), failure to return to usual activities (*blue*), failure to return to usual health (*yellow*), and Long COVID (*green*).** Whiskers representing 95% confidence limits.

vaccine receipt, and sampling time ($P < 0.01$). Participants who were obese and severely obese also had higher CRP and IL-6 levels compared with those with normal weight, after adjusting for sampling time ($P < 0.01$) (S8, S9 Tables in S1 File). The participants in the Respiratory/ Systemic cluster have higher CRP and IL-6 levels than those in the Nasal and Sensory clusters ($P < 0.01$) (Fig 3).

## Discussion

Distinguishing and interpreting patterns of acute symptoms of COVID-19 is inherently challenging. In this study, we used machine learning techniques to identify groups with distinct early symptom profile clusters. Our results indicate that participants with early COVID-19 symptoms belonging to the Respiratory/Systemic cluster were more likely to be hospitalized than those in the Sensory and Nasal early-symptom clusters. The Respiratory/Systemic cluster

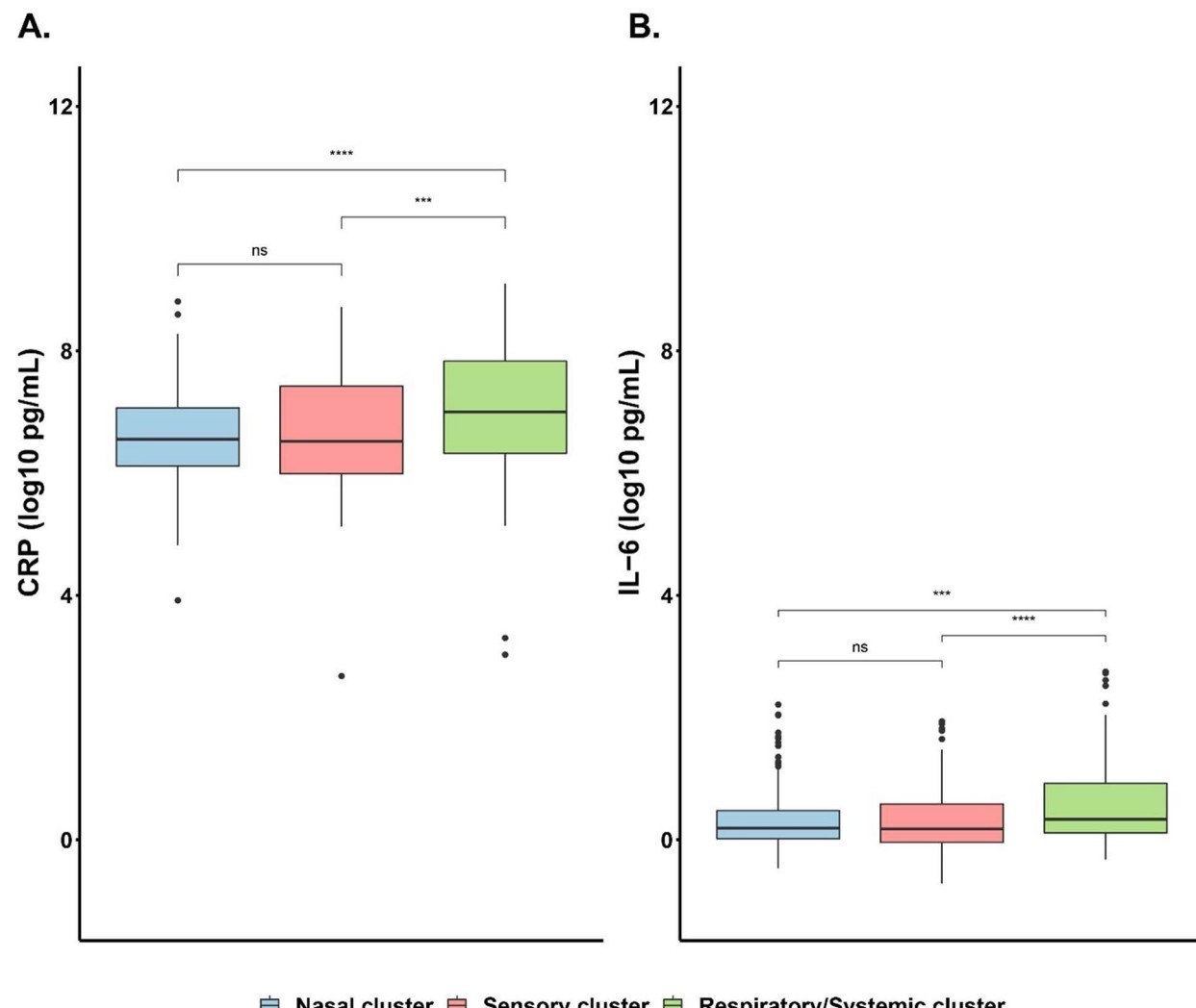

**Fig 3.** Comparison of inflammatory biomarker (A) CRP and (B) IL-6 in identified clusters. Statistical significance was determined by Wilcoxon rank sum test. Asterisks indicate statistical significance: ns: p > 0.05, *: p ≤ 0.05, **: p ≤ 0.01, ***: p ≤ 0.001, ****: p ≤ 0.0001.

was also independently associated with older age, more comorbidities, and obesity, factors which have been found to be associated with increased COVID-19 severity in the EPICC cohort and other studies [30, 35–39]. Participants in the Respiratory/Systemic cluster were also more likely to report that they had not yet returned to usual health and activities at the end of the FLU-PRO Plus survey (Fig 2).

With growing concern about long-term symptoms associated with SARS-CoV-2 infection, and uncertainty on how to predict long term symptom persistence/PCC, we performed sub-analyses in participants who completed a (non-FLU-PRO) symptom survey at six months post-symptom onset. Cases in the early FLU-PRO Sensory and Respiratory/Systemic clusters reported more prolonged COVID-19 symptoms at six months than those in the Nasal cluster.

The mechanism for such longer term COVID-19 symptoms, and how acute symptom profiles predict such late sequelae, is unclear [40–44]. One hypothesis may be that these late post-COVID-19 manifestations are associated with prolonged inflammation. Indeed, CRP and IL-6 serum levels were higher in participants in the Respiratory/Systemic cluster than those in the Sensory and Nasal clusters after adjusting for age, race/ethnicity, sex and sampling time, suggesting greater systemic inflammation in these patients. This would be consistent with the Respiratory/Systemic cluster also correlating with hospitalization risk, which in turn is well known to be associated with higher IL-6 and CRP [45, 46]. Additionally, acute presentation with predominantly nasal ("cold-like") symptoms may represent less invasive and less severe disease in turn connoting a decreased risk of both severe COVID and Long COVID.

Our findings shows that the Nasal cluster type symptoms were more prominent among those infected with the Omicron (BA.1/BA.2) variant, consistent with recent studies suggesting that Omicron (BA.1) may have a greater tropism for the upper respiratory tract and putatively lower virulence [47] compared with prior variants [48, 49]. Omicron has also been found to replicate 70 times faster than Delta in the large bronchi but replicates ten times slower in lung parenchyma than the ancestral variant [50, 51].

This analysis has several caveats and prompts further study. First, given the subjectivity of symptom measurement (even with the standardized FLU-PRO scoring system) and given that only a subset of those in our cohort filled out six-month surveys (because long term follow-up is ongoing for more recent enrollees) (S1, S5 Tables in S1 File), our findings should be cross validated in separate cohorts from other populations. Second, these findings are limited to statistical associations and limited inflammatory profiling; further mechanistic research (e.g., transcriptomic data showing differential SARS-CoV-2 receptor expression data among those in the Nasal cluster) may help describe the pathophysiology behind these distinct clusters and their association with short- and long-term outcomes.

The strengths of this study include the use of a standardized measurement tool (FLU-PRO Plus) which quantifies respiratory infection symptom severity as part of prospective data collection. In addition, we utilized unsupervised machine learning technique to visualize patterns in symptom data, which allowed for the identification of symptom clusters in a large cohort where such distinct patterns were not otherwise apparent.

Taken together, these findings suggest distinct COVID-19 symptom profiles are associated with differential short- and long-term outcomes and may help improve COVID-19 prognostication. Our further delineation of inflammatory profiles associated with these acute symptom clusters may further assist in understanding the mechanism of developing long term post-COVID complications and may direct further study into Long COVID prevention and treatment.

## Supporting information

**S1 File.**
(DOCX)

## Acknowledgments

We sincerely thank the members of the EPICC COVID-19 Cohort Study Group for their many contributions in conducting the study and ensuring effective protocol operations. The following members were all closely involved with the design, implementation, and oversight of the study and have met group authorship criteria for this manuscript:

*Brooke Army Medical Center, Fort Sam Houston, TX*: Jessica J. Cowden; Teresa M. Merritt

*Fort Belvoir Community Hospital, Fort Belvoir, VA*: Nora Elnahas; Christa Glinn; Donna Jennings; Chiquita West

*ACESO, Henry M. Jackson Foundation, Inc., Bethesda, MD*: Danielle Clark

*Madigan Army Medical Center, Joint Base Lewis McChord, WA*: Susan Chambers; Cristin A. Mount

*Naval Medical Center San Diego, San Diego, CA*: Nichol M. Kirkland

*Tripler Army Medical Center, Honolulu, HI*: Catherine Uyehara

*Uniformed Services University of the Health Sciences, Bethesda, MD*: Heidi Adams; Celia Byrne; Mark Fritschlanski; Edward Parmelee; Jennifer Rusiecki; Emily Samuels; Ann Scher; Melissa Wayman

*United States Air Force School of Aerospace Medicine, Dayton, OH*: Richard Chapleau; Monica Christian; Kelsey Lanter; Elizabeth Macias

*United States Coast Guard, Washington, DC*: John K. Iskander

*Womack Army Medical Center, Fort Bragg, NC*: Kathryn J. Lago

The authors wish to also acknowledge all who have contributed to the EPICC COVID-19 study:

*Brooke Army Medical Center, Fort Sam Houston, TX*: J. Cowden; M. Darling; S. DeLeon; D. Lindholm; A. Markelz; K. Mende; S. Merritt; T. Merritt; N. Turner; T. Wellington

*Carl R. Darnall Army Medical Center, Fort Hood, TX*: S. Bazan; D. Hrncir; P.K Love

*Fort Belvoir Community Hospital, Fort Belvoir, VA*: N. Dimascio-Johnson; N. Elnahas; E. Ewers; K. Gallagher; C. Glinn; U. Jarral; D. Jennings; D. Larson; A. Mentzos; K. Reterstoff; A. Rutt; A. Silva; C. West

*Henry M. Jackson Foundation, Inc., Bethesda, MD*: P. Blair; J. Chenoweth; D. Clark

*Madigan Army Medical Center, Joint Base Lewis McChord, WA*: J. Bowman; S. Chambers; C. Colombo; R. Colombo; C. Conlon; K. Everson; P. Faestel; T. Ferguson; L. Gordon; S. Grogan; S. Lis; M. Martin; C. Mount; D. Musfeldt; D. Odineal; M. Perreault; W. Robb-McGrath; R. Sainato; C. Schofield; C. Skinner; M. Stein; M. Switzer; M. Timlin; S. Wood

*Naval Medical Center Portsmouth, Portsmouth, VA*: S. Banks; R. Carpenter; L. Kim; K. Kronmann; T. Lalani; T. Lee; A. Smith; R. Smith; R. Tant; T. Warkentien

*Naval Medical Center San Diego, San Diego, CA*: C. Berjohn; S. Cammarata; N. Kirkland; D. Libraty; R. Maves; G. Utz

*Tripler Army Medical Center, Honolulu, HI*: C. Bradley; S. Chi; R. Flanagan; A. Fuentes; M. Jones; N. Leslie; C. Lucas; C. Madar; K. Miyasato; C. Uyehara

*Uniformed Services University of the Health Sciences, Bethesda, MD*: H. Adams; B. Agan; L. Andronescu; A. Austin; C. Broder; T. Burgess; C. Byrne; K Chung; J. Davies; C. English; N. Epsi; C. Fox; M. Fritschlanski; A. Hadley; P. Hickey; E. Laing; C. Lanteri; J. Livezey; A. Malloy; R. Mohammed; C. Morales; P. Nwachukwu; C. Olsen; E. Parmelee; S. Pollett; S. Richard; J. Rozman; J. Rusiecki; D. Saunders; E. Samuels; M. Sanchez; A. Scher; M. Simons; A. Snow; K. Telu; D. Tribble; M. Tso; L. Ulomi; M. Wayman

*United States Air Force School of Aerospace Medicine, Dayton, OH:* T. Chao; R. Chapleau; M. Christian; A. Fries; C. Harrington; V. Hogan; S. Huntsberger; K. Lanter; E. Macias; J. Meyer; S. Purves; K. Reynolds; J. Rodriguez; C. Starr

*United States Coast Guard*, *Washington*, *DC*: J. Iskander; I. Kamara

*Womack Army Medical Center*, *Fort Bragg*, *NC*: B. Barton; D. Hostler; J. Hostler; K. Lago; C. Maldonado; J. Mehrer

*William Beaumont Army Medical Center*, *El Paso*, *TX*: T. Hunter; J. Mejia; R. Mody; J. Montes; R. Resendez; P. Sandoval

*Walter Reed National Military Medical Center*, *Bethesda*, *MD*: I. Barahona; A. Baya; A. Ganesan; N. Huprikar; B. Johnson

**Disclaimer:** The contents of this publication are the sole responsibility of the author (s) and do not necessarily reflect the views, opinions, or policies of Uniformed Services University of the Health Sciences (USUHS); the Department of Defense (DoD); the Departments of the Army, Navy, or Air Force; the Defense Health Agency, Brooke Army Medical Center; Walter Reed National Military Medical Center; Naval Medical Center San Diego; Madigan Army Medical Center; United States Air Force School of Aerospace Medicine; Fort Belvoir Community Hospital; Carl R. Darnall Army Medical Center; Naval Medical Center Portsmouth, Portsmouth, VA; Tripler Army Medical Center, Honolulu, HI; United States Coast Guard, Washington, DC; Womack Army Medical Center, Fort Bragg, NC; William Beaumont Army Medical Center, El Paso, TX: the Henry M. Jackson Foundation for the Advancement of Military Medicine Inc; the National Institutes of Health. Mention of trade names, commercial products, or organizations does not imply endorsement by the U.S. Government. The investigators have adhered to the policies for protection of human subjects as prescribed in 45 CFR 46.

## Author contributions

**Conceptualization:** Stephanie A. Richard.

**Data curation:** Nusrat J. Epsi.

**Formal analysis:** Stephanie A. Richard.

**Funding acquisition:** Mark P. Simons, David R. Tribble, Brian K. Agan, Timothy H. Burgess.

**Investigation:** Nusrat J. Epsi, Simon D. Pollett, Stephanie A. Richard.

**Methodology:** Nusrat J. Epsi, Simon D. Pollett, Stephanie A. Richard.

**Project administration:** Julia S. Rozman, Mark P. Simons, David R. Tribble, Brian K. Agan, Timothy H. Burgess, Simon D. Pollett, Stephanie A. Richard.

**Resources:** Nusrat J. Epsi, Katrin Mende, Allison Malloy, Anuradha Ganesan, Nikhil Huprikar, Tahaniyat Lalani, Alfred Smith, Rupal M. Mody, Milissa U. Jones, Samantha E. Bazan, Rhonda E. Colombo, Christopher J. Colombo, Evan C. Ewers, Derek T. Larson, Catherine M. Berjohn, Carlos J. Maldonado, Paul W. Blair, Josh Chenoweth, David L. Saunders, Jeffrey Livezey, Ryan C. Maves, Margaret Sanchez Edwards, Simon D. Pollett, Stephanie A. Richard.

**Software:** Nusrat J. Epsi.

**Supervision:** Nusrat J. Epsi, John H. Powers, David A. Lindholm, Mark P. Simons, Simon D. Pollett, Stephanie A. Richard.

**Validation:** Nusrat J. Epsi.

**Visualization:** Nusrat J. Epsi.

**Writing – original draft:** Nusrat J. Epsi, Stephanie A. Richard.

**Writing – review & editing:** Nusrat J. Epsi, Simon D. Pollett, Stephanie A. Richard.

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
