## [Decision Letter · Decision Letter 0]

19 Dec 2022

PONE-D-22-31484A machine learning approach identifies distinct early-symptom cluster phenotypes which correlate with hospitalization, failure to return to activities, and prolonged COVID-19 symptomsPLOS ONEPONE-D-22-31484A machine learning approach identifies distinct early-symptom cluster phenotypes which correlate with hospitalization, failure to return to activities, and prolonged COVID-19 symptomsPLOS ONE

Dear Dr. Richard, Dear Dr. Richard,

Thank you for submitting your manuscript to PLOS ONE. After careful consideration, we feel that it has merit but does not fully meet PLOS ONE’s publication criteria as it currently stands. Therefore, we invite you to submit a revised version of the manuscript that addresses the points raised during the review process.Thank you for submitting your manuscript to PLOS ONE. After careful consideration, we feel that it has merit but does not fully meet PLOS ONE’s publication criteria as it currently stands. Therefore, we invite you to submit a revised version of the manuscript that addresses the points raised during the review process.

Please include the following items when submitting your revised manuscript:Please include the following items when submitting your revised manuscript:A rebuttal letter that responds to each point raised by the academic editor and reviewer(s). You should upload this letter as a separate file labeled 'Response to Reviewers'.A rebuttal letter that responds to each point raised by the academic editor and reviewer(s). You should upload this letter as a separate file labeled 'Response to Reviewers'.A marked-up copy of your manuscript that highlights changes made to the original version. You should upload this as a separate file labeled 'Revised Manuscript with Track Changes'.A marked-up copy of your manuscript that highlights changes made to the original version. You should upload this as a separate file labeled 'Revised Manuscript with Track Changes'.An unmarked version of your revised paper without tracked changes. You should upload this as a separate file labeled 'Manuscript'.An unmarked version of your revised paper without tracked changes. You should upload this as a separate file labeled 'Manuscript'.We look forward to receiving your revised manuscript.We look forward to receiving your revised manuscript.

Kind regards,Kind regards,

Dr Gayathri DelanerolleDr Gayathri Delanerolle

Academic EditorAcademic Editor

PLOS ONEPLOS ONE

Journal Requirements:Journal Requirements:

When submitting your revision, we need you to address these additional requirements.When submitting your revision, we need you to address these additional requirements.

1. Please ensure that your manuscript meets PLOS ONE's style requirements, including those for file naming. The PLOS ONE style templates can be found at 1. Please ensure that your manuscript meets PLOS ONE's style requirements, including those for file naming. The PLOS ONE style templates can be found at

https://journals.plos.org/plosone/s/file?id=wjVg/PLOSOne_formatting_sample_main_body.pdf and and

2. Please amend your current ethics statement to address the following concerns:2. Please amend your current ethics statement to address the following concerns:

a) Did participants provide their written or verbal informed consent to participate in this study?a) Did participants provide their written or verbal informed consent to participate in this study?

b) If consent was verbal, please explain i) why written consent was not obtained, ii) how you documented participant consent, and iii) whether the ethics committees/IRB approved this consent procedure.b) If consent was verbal, please explain i) why written consent was not obtained, ii) how you documented participant consent, and iii) whether the ethics committees/IRB approved this consent procedure.

3. Thank you for stating the following in the Competing Interests section: 3. Thank you for stating the following in the Competing Interests section:

"Potential conflicts of interest. S. D. P., T. H. B, and M.P.S. report that the Uniformed Services University (USU) Infectious Diseases Clinical Research Program (IDCRP), a US Department of Defense institution, and the Henry M. Jackson Foundation (HJF) were funded under a Cooperative Research and Development Agreement to conduct an unrelated phase III COVID-19 monoclonal antibody immunoprophylaxis trial sponsored by AstraZeneca. The HJF, in support of the USU IDCRP, was funded by the Department of Defense Joint Program Executive Office for Chemical, Biological, Radiological, and Nuclear Defense to augment the conduct of an unrelated phase III vaccine trial sponsored by AstraZeneca. Both of these trials were part of the US Government COVID-19 response. Neither is related to the work presented here""Potential conflicts of interest. S. D. P., T. H. B, and M.P.S. report that the Uniformed Services University (USU) Infectious Diseases Clinical Research Program (IDCRP), a US Department of Defense institution, and the Henry M. Jackson Foundation (HJF) were funded under a Cooperative Research and Development Agreement to conduct an unrelated phase III COVID-19 monoclonal antibody immunoprophylaxis trial sponsored by AstraZeneca. The HJF, in support of the USU IDCRP, was funded by the Department of Defense Joint Program Executive Office for Chemical, Biological, Radiological, and Nuclear Defense to augment the conduct of an unrelated phase III vaccine trial sponsored by AstraZeneca. Both of these trials were part of the US Government COVID-19 response. Neither is related to the work presented here"

Please confirm that this does not alter your adherence to all PLOS ONE policies on sharing data and materials, by including the following statement: "This does not alter our adherence to  PLOS ONE policies on sharing data and materials.” (as detailed online in our guide for authors Please confirm that this does not alter your adherence to all PLOS ONE policies on sharing data and materials, by including the following statement: "This does not alter our adherence to  PLOS ONE policies on sharing data and materials.” (as detailed online in our guide for authors http://journals.plos.org/plosone/s/competing-interests). If there are restrictions on sharing of data and/or materials, please state these. Please note that we cannot proceed with consideration of your article until this information has been declared. ).  If there are restrictions on sharing of data and/or materials, please state these. Please note that we cannot proceed with consideration of your article until this information has been declared.

Please include your updated Competing Interests statement in your cover letter; we will change the online submission form on your behalf.Please include your updated Competing Interests statement in your cover letter; we will change the online submission form on your behalf.

4. In your Data Availability statement, you have not specified where the minimal data set underlying the results described in your manuscript can be found. PLOS defines a study's minimal data set as the underlying data used to reach the conclusions drawn in the manuscript and any additional data required to replicate the reported study findings in their entirety. All PLOS journals require that the minimal data set be made fully available. For more information about our data policy, please see 4. In your Data Availability statement, you have not specified where the minimal data set underlying the results described in your manuscript can be found. PLOS defines a study's minimal data set as the underlying data used to reach the conclusions drawn in the manuscript and any additional data required to replicate the reported study findings in their entirety. All PLOS journals require that the minimal data set be made fully available. For more information about our data policy, please see http://journals.plos.org/plosone/s/data-availability.

""Upon re-submitting your revised manuscript, please upload your study’s minimal underlying data set as either Supporting Information files or to a stable, public repository and include the relevant URLs, DOIs, or accession numbers within your revised cover letter. For a list of acceptable repositories, please see ""Upon re-submitting your revised manuscript, please upload your study’s minimal underlying data set as either Supporting Information files or to a stable, public repository and include the relevant URLs, DOIs, or accession numbers within your revised cover letter. For a list of acceptable repositories, please see http://journals.plos.org/plosone/s/data-availability#loc-recommended-repositories. Any potentially identifying patient information must be fully anonymized.. Any potentially identifying patient information must be fully anonymized.

Important: If there are ethical or legal restrictions to sharing your data publicly, please explain these restrictions in detail. Please see our guidelines for more information on what we consider unacceptable restrictions to publicly sharing data: Important: If there are ethical or legal restrictions to sharing your data publicly, please explain these restrictions in detail. Please see our guidelines for more information on what we consider unacceptable restrictions to publicly sharing data: http://journals.plos.org/plosone/s/data-availability#loc-unacceptable-data-access-restrictions. Note that it is not acceptable for the authors to be the sole named individuals responsible for ensuring data access.. Note that it is not acceptable for the authors to be the sole named individuals responsible for ensuring data access.

We will update your Data Availability statement to reflect the information you provide in your cover letter.We will update your Data Availability statement to reflect the information you provide in your cover letter.

5. One of the noted authors is a group or consortium "EPICC COVID-19 Cohort Study Group". In addition to naming the author group, please list the individual authors and affiliations within this group in the acknowledgments section of your manuscript. Please also indicate clearly a lead author for this group along with a contact email address.5. One of the noted authors is a group or consortium "EPICC COVID-19 Cohort Study Group". In addition to naming the author group, please list the individual authors and affiliations within this group in the acknowledgments section of your manuscript. Please also indicate clearly a lead author for this group along with a contact email address.

6. Please include captions for your Supporting Information files at the end of your manuscript, and update any in-text citations to match accordingly. Please see our Supporting Information guidelines for more information: 6. Please include captions for your Supporting Information files at the end of your manuscript, and update any in-text citations to match accordingly. Please see our Supporting Information guidelines for more information: http://journals.plos.org/plosone/s/supporting-information..

7. Please review your reference list to ensure that it is complete and correct. If you have cited papers that have been retracted, please include the rationale for doing so in the manuscript text, or remove these references and replace them with relevant current references. Any changes to the reference list should be mentioned in the rebuttal letter that accompanies your revised manuscript. If you need to cite a retracted article, indicate the article’s retracted status in the References list and also include a citation and full reference for the retraction notice.7. Please review your reference list to ensure that it is complete and correct. If you have cited papers that have been retracted, please include the rationale for doing so in the manuscript text, or remove these references and replace them with relevant current references. Any changes to the reference list should be mentioned in the rebuttal letter that accompanies your revised manuscript. If you need to cite a retracted article, indicate the article’s retracted status in the References list and also include a citation and full reference for the retraction notice.

Reviewers' comments:Reviewers' comments:

Reviewer's Responses to QuestionsReviewer's Responses to Questions

**Comments to the Author**

1. Is the manuscript technically sound, and do the data support the conclusions?1. Is the manuscript technically sound, and do the data support the conclusions?

The manuscript must describe a technically sound piece of scientific research with data that supports the conclusions. Experiments must have been conducted rigorously, with appropriate controls, replication, and sample sizes. The conclusions must be drawn appropriately based on the data presented. The manuscript must describe a technically sound piece of scientific research with data that supports the conclusions. Experiments must have been conducted rigorously, with appropriate controls, replication, and sample sizes. The conclusions must be drawn appropriately based on the data presented.

Reviewer #1: YesReviewer #1: Yes

2. Has the statistical analysis been performed appropriately and rigorously? 2. Has the statistical analysis been performed appropriately and rigorously?

Reviewer #1: YesReviewer #1: Yes

3. Have the authors made all data underlying the findings in their manuscript fully available?3. Have the authors made all data underlying the findings in their manuscript fully available?

The The PLOS Data policy requires authors to make all data underlying the findings described in their manuscript fully available without restriction, with rare exception (please refer to the Data Availability Statement in the manuscript PDF file). The data should be provided as part of the manuscript or its supporting information, or deposited to a public repository. For example, in addition to summary statistics, the data points behind means, medians and variance measures should be available. If there are restrictions on publicly sharing data—e.g. participant privacy or use of data from a third party—those must be specified.requires authors to make all data underlying the findings described in their manuscript fully available without restriction, with rare exception (please refer to the Data Availability Statement in the manuscript PDF file). The data should be provided as part of the manuscript or its supporting information, or deposited to a public repository. For example, in addition to summary statistics, the data points behind means, medians and variance measures should be available. If there are restrictions on publicly sharing data—e.g. participant privacy or use of data from a third party—those must be specified.

Reviewer #1: NoReviewer #1: No

4. Is the manuscript presented in an intelligible fashion and written in standard English?4. Is the manuscript presented in an intelligible fashion and written in standard English?

PLOS ONE does not copyedit accepted manuscripts, so the language in submitted articles must be clear, correct, and unambiguous. Any typographical or grammatical errors should be corrected at revision, so please note any specific errors here.PLOS ONE does not copyedit accepted manuscripts, so the language in submitted articles must be clear, correct, and unambiguous. Any typographical or grammatical errors should be corrected at revision, so please note any specific errors here.

Reviewer #1: YesReviewer #1: Yes

5. Review Comments to the Author5. Review Comments to the Author

Please use the space provided to explain your answers to the questions above. You may also include additional comments for the author, including concerns about dual publication, research ethics, or publication ethics. (Please upload your review as an attachment if it exceeds 20,000 characters)Please use the space provided to explain your answers to the questions above. You may also include additional comments for the author, including concerns about dual publication, research ethics, or publication ethics. (Please upload your review as an attachment if it exceeds 20,000 characters)

Reviewer #1: Authors have presented an important piece of information in understanding symptom patterns of the Covid-19 and likely clinical course. I have few suggestions for consideration of the authorsReviewer #1: Authors have presented an important piece of information in understanding symptom patterns of the Covid-19 and likely clinical course. I have few suggestions for consideration of the authors

1. It will be helpful for international readers if it is clarified that the beneficiaries included/not included the dependent family members of the military. Also, in the table 1, the occupation might be helpful, as a predominant in-service participants might have a different level of pre-illness physical fitness1. It will be helpful for international readers if it is clarified that the beneficiaries included/not included the dependent family members of the military. Also, in the table 1, the occupation might be helpful, as a predominant in-service participants might have a different level of pre-illness physical fitness

2. It is also noted that only half of the enrolled participants completed FLU-PRO Plus survey, and further around half of these participants responded to the survey at six month. Therefore, authors may present some demographic of all enrolled participants and clinical characteristics of participants who completed the first survey, for benefit of readers to be able to compare the participating and non-participating enrollees. This may be added as supplement.2. It is also noted that only half of the enrolled participants completed FLU-PRO Plus survey, and further around half of these participants responded to the survey at six month. Therefore, authors may present some demographic of all enrolled participants and clinical characteristics of participants who completed the first survey, for benefit of readers to be able to compare the participating and non-participating enrollees. This may be added as supplement.

6. PLOS authors have the option to publish the peer review history of their article (6. PLOS authors have the option to publish the peer review history of their article (what does this mean?). If published, this will include your full peer review and any attached files.). If published, this will include your full peer review and any attached files.

.

Reviewer #1: NoReviewer #1: No

---

## [Author Response · Author response to Decision Letter 0]

5 Jan 2023

Responses to the editor and reviewer were uploaded separatelyResponses to the editor and reviewer were uploaded separately

---

## [Editor Report · Decision Letter 1]

19 Jan 2023

A machine learning approach identifies distinct early-symptom cluster phenotypes which correlate with hospitalization, failure to return to activities, and prolonged COVID-19 symptomsA machine learning approach identifies distinct early-symptom cluster phenotypes which correlate with hospitalization, failure to return to activities, and prolonged COVID-19 symptoms

PONE-D-22-31484R1PONE-D-22-31484R1

Dear Dr Richard,Dear Dr Richard,

We’re pleased to inform you that your manuscript has been judged scientifically suitable for publication and will be formally accepted for publication once it meets all outstanding technical requirements.We’re pleased to inform you that your manuscript has been judged scientifically suitable for publication and will be formally accepted for publication once it meets all outstanding technical requirements.

Within one week, you’ll receive an e-mail detailing the required amendments. When these have been addressed, you’ll receive a formal acceptance letter and your manuscript will be scheduled for publication.Within one week, you’ll receive an e-mail detailing the required amendments. When these have been addressed, you’ll receive a formal acceptance letter and your manuscript will be scheduled for publication.

An invoice for payment will follow shortly after the formal acceptance. To ensure an efficient process, please log into Editorial Manager at An invoice for payment will follow shortly after the formal acceptance. To ensure an efficient process, please log into Editorial Manager at http://www.editorialmanager.com/pone/, click the 'Update My Information' link at the top of the page, and double check that your user information is up-to-date. If you have any billing related questions, please contact our Author Billing department directly at , click the 'Update My Information' link at the top of the page, and double check that your user information is up-to-date. If you have any billing related questions, please contact our Author Billing department directly at authorbilling@plos.org..

If your institution or institutions have a press office, please notify them about your upcoming paper to help maximize its impact. If they’ll be preparing press materials, please inform our press team as soon as possible -- no later than 48 hours after receiving the formal acceptance. Your manuscript will remain under strict press embargo until 2 pm Eastern Time on the date of publication. For more information, please contact If your institution or institutions have a press office, please notify them about your upcoming paper to help maximize its impact. If they’ll be preparing press materials, please inform our press team as soon as possible -- no later than 48 hours after receiving the formal acceptance. Your manuscript will remain under strict press embargo until 2 pm Eastern Time on the date of publication. For more information, please contact onepress@plos.org..

Kind regards,Kind regards,

Dr Gayathri DelanerolleDr Gayathri Delanerolle

Academic EditorAcademic Editor

PLOS ONEPLOS ONE

---

## [Editor Report · Acceptance letter]

30 Jan 2023

PONE-D-22-31484R1

PONE-D-22-31484R1

A machine learning approach identifies distinct early-symptom cluster phenotypes which correlate with hospitalization, failure to return to activities, and prolonged COVID-19 symptoms   A machine learning approach identifies distinct early-symptom cluster phenotypes which correlate with hospitalization, failure to return to activities, and prolonged COVID-19 symptoms

Dear Dr. Richard:Dear Dr. Richard:

I'm pleased to inform you that your manuscript has been deemed suitable for publication in PLOS ONE. Congratulations! Your manuscript is now with our production department. I'm pleased to inform you that your manuscript has been deemed suitable for publication in PLOS ONE. Congratulations! Your manuscript is now with our production department.

If your institution or institutions have a press office, please let them know about your upcoming paper now to help maximize its impact. If they'll be preparing press materials, please inform our press team within the next 48 hours. Your manuscript will remain under strict press embargo until 2 pm Eastern Time on the date of publication. For more information please contact If your institution or institutions have a press office, please let them know about your upcoming paper now to help maximize its impact. If they'll be preparing press materials, please inform our press team within the next 48 hours. Your manuscript will remain under strict press embargo until 2 pm Eastern Time on the date of publication. For more information please contact onepress@plos.org..

If we can help with anything else, please email us at If we can help with anything else, please email us at plosone@plos.org. .

Thank you for submitting your work to PLOS ONE and supporting open access. Thank you for submitting your work to PLOS ONE and supporting open access.

Kind regards, Kind regards,

PLOS ONE Editorial Office StaffPLOS ONE Editorial Office Staff

on behalf ofon behalf of

Dr. Gayathri Delanerolle Dr. Gayathri Delanerolle

Academic EditorAcademic Editor

PLOS ONEPLOS ONE